# Kinetic Evaluation of Dye Decolorization by Fenton Processes in the Presence of 3-Hydroxyanthranilic Acid

**DOI:** 10.3390/ijerph16091602

**Published:** 2019-05-07

**Authors:** Cássia Sidney Santana, Márcio Daniel Nicodemos Ramos, Camila Cristina Vieira Velloso, André Aguiar

**Affiliations:** 1Campus Alto Paraopeba, Universidade Federal de São João Del-Rei, Ouro Branco 36420-000, MG, Brazil; kciasantana_02@hotmail.com (C.S.S.); camila.cvelloso@gmail.com (C.C.V.V.); 2Instituto de Recursos Naturais, Universidade Federal de Itajubá, Itajubá 37500-903, MG, Brazil; marcio_daniel_ramos@hotmail.com

**Keywords:** Fenton reaction, 3-Hydroxyanthranilic, Dye decolorization, Kinetics, Pro-oxidant properties, iron

## Abstract

The fungal metabolite 3-hydroxyanthranilic acid (3-HAA) was used as a redox mediator with the aim of increasing dye degradation by Fenton oxidative processes (Fe^2+^/H_2_O_2_, Fe^3+^/H_2_O_2_). Its Fe^3+^-reducing activity can enhance the generation of reactive oxygen species as HO^●^ radicals. Initially, the influence of 3-HAA on decolorization kinetics of five dyes (methylene blue, chromotrope 2R, methyl orange, phenol red, and safranin T) was investigated using decolorization data from a previous work conducted by the present research group. Fe^3+^-containing reaction data were well fitted with first-order and mainly second-order kinetic models, whereas the BMG (Behnajady, Modirshahla and Ghanbary) model obtained optimal fit to Fe^2+^. Improvements in kinetic parameters (i.e., apparent rate constants and maximum oxidation capacity) were observed with the addition of 3-HAA. In another set of experiments, a decrease in apparent activation energy was observed due to introducing 3-HAA into reactions containing either Fe^2+^ or Fe^3+^ in order to decolorize phenol red at different temperatures. This indicates that the redox mediator decreases the energy barrier so as to allow reactions to occur. Thus, based on recent experiments and the reaction kinetics models evaluated herein, pro-oxidant properties have been observed for 3-HAA in Fenton processes.

## 1. Introduction

One of the major problems faced nowadays is water pollution caused by different industrial activities, which can raise thorny environmental, economic, social, and health issues [1,2]. In this scenario, the textile industry should be highlighted, due to the fact that it is accountable for generating massive amounts of effluent containing recalcitrant dyes and other pollutants. Textile wastewater must be treated so as to remove or decrease the concentration of pollutants to acceptable levels before its reuse or being discharged into the environment [3]. The biggest challenge is that many textile dyes are toxic and unable to be degraded by conventional biological treatments [3,4]. Due to such limitations, new alternatives are being sought in order to overcome the inefficiency of conventional techniques for removal or degradation of dyes. Advanced oxidative processes (AOPs) have demonstrated great efficiency in degrading recalcitrant organic pollutants [5,6,7,8,9,10,11,12]. Among AOPs, those based on Fenton reactions are considered promising for dye degradation [13].

The Fenton reaction generates reactive oxygen species with great oxidation potential, such as the hydroxyl radical (HO^●^) from hydrogen peroxide by using Fe^2+^ as catalyst (Equation (1)). This is referred to as the classical Fenton reaction. These radicals are capable of oxidizing different organic pollutants into simpler molecules or converting them into CO_2_, H_2_O, and inorganic ions. Fe^3+^ can also be used as a catalyst, since it is more abundant; its reaction is otherwise known as Fenton-like reaction (Equation (2)). However, the latter is slower and generates an oxyhydroxyl radical (HO_2_^●^) with an oxidation capacity that is far inferior to HO^●^. On the other hand, it is important to reduce Fe^3+^ to Fe^2+^ so that it reacts with H_2_O_2_ [5,9,10]. In order to increase the generation of HO^●^, regardless of the initial oxidation state of iron, the use of Fe^2+^-regenerating compounds is a promising strategy in Fenton processes [14].
Fe^2+^ + H_2_O_2_ → Fe^3+^ + HO^●^ + ^−^OH *k* = 50–80 mol^−^^1^ L s^−^^1^(1)
Fe^3+^ + H_2_O_2_ → Fe^2+^ + HO_2_^●^ + H^+^*k* = 0.002–0.01 mol^−1^ L s^−1^(2)

Different compounds that rapidly and continuously reduce Fe^3+^ to Fe^2+^ have been extensively studied. They can be either synthetic or of a natural origin. Among synthetic compounds, dihydroxybenzenes are those that have been the most widely evaluated in literature [15,16,17,18,19]. Natural compounds, such as the amino acid cysteine [20,21], ascorbic acid [22,23], phenols derived from lignin such as vanillin [24,25], and gallic acid derived from tannins [19,26,27] have also caused an increase in the degradation of dyes and other organic pollutants by Fenton processes. 3-Hydroxyanthranilic acid (3-HAA; Figure 1) is another important and well-known Fe^3+^ reducer, which is produced by the fungi *Pycnoporus cinnabarinus* [28] and *Criptococcus neoformans* [29,30]. The pro-oxidant properties of this compound were first verified through decolorizing azure B dye by Fe^3+^/H_2_O_2_ and Cu^2+^/H_2_O_2_ [31].

In a more recent study, improvements were also observed in the decolorization of other dyes by Fe^3+^/H_2_O_2_ and Fe^2+^/H_2_O_2_ systems due to the addition of 3-HAA as redox mediator to the reactions [32]. Nevertheless, it is necessary to evaluate how this fungal metabolite can influence the rate of dye decolorization based on different kinetic models. As a matter of fact, kinetic models obtained on a laboratorial scale are crucial to allow reactor design, scale-up, and performance prediction. A few recently published kinetics studies have only made use of a first-order linear model to verify the pro-oxidant effect of redox mediators through Fenton processes. However, it can be observed that it is not always the most appropriate, since some correlation coefficients describing the kinetics of reactions (R^2^) were lower than 0.9 [20,22,33]. Its lack of fit may compromise the analysis and hinder a comparison of different reaction systems. Thus, the present work aimed to perform a kinetic study based on different models (zero-, first-, second-order, and BMG—Behnajady, Modirshahla, and Ghanbary), using data from a previous work carried out by this research group, in which the decolorization of five dyes treated by homogeneous Fenton systems (Fe^2+^/H_2_O_2_, Fe^3+^/H_2_O_2_) was evaluated in both the absence and presence of 3-HAA [32]. In addition to determining reaction rate constants, another set of experiments was carried out herein to verify the effect of temperature on the determination of activation energy as a function of the addition of 3-HAA to these reactions.

## 2. Materials and Methods

### 2.1. Chemicals

Chromotrope 2R and 3-hydroxyanthranilic acid were supplied by Sigma-Aldrich Chemical Co. (Milwaukee WI, USA); Fe(NO_3_)_3_, methyl orange, safranin T, phenol red, and H_2_O_2_ (30% m/m) were purchased from Vetec (Rio de Janeiro, Brazil); FeSO_4_ was obtained from Synth (São Paulo, Brazil). Other reagents were of analytical grade from several suppliers. All reagents were used without prior purification. Deionized water was used to prepare all solutions. Table 1 summarizes some of the features of the dyes evaluated herein.

### 2.2. Decolorization of Dyes

In order to evaluate the decolorization of dyes, reactions were performed inside 3 mL quartz cuvettes containing 30 µmol L^−1^ dye, 450 µmol L^−1^ H_2_O_2_, 30 µmol L^−1^ freshly prepared FeSO_4_ or Fe(NO_3_)_3_ using a classical Fenton reaction and a Fenton-like reaction, respectively, and 1 mmol L^−1^ H_2_SO_4_ so as to adjust the pH between 2.5 and 3.0. As regards reactions involving the mediator, an amount of stock solution was added to reach a final concentration of 10 µmol L^−1^ 3-HAA. Deionized water was added in order to reach a final volume of 2 mL. Fe ion solutions were added to the mixture last so as to trigger the reactions. Decolorization was expressed as a decrease in absorbance of the wavelength at which each dye presents maximum absorbance (λ_max_, Table 1). This set of analyses was performed in duplicate using a UV/vis spectrophotometer (Biochrom Libra S50). Dye decolorization was monitored at 5, 10, 20, 40, and 60 min after the reactions had started. The blank was prepared in the absence of dyes and iron salts. All experiments were performed at room temperature in the dark and under static conditions. Further details about the experimental conditions have been reported by Santana and Aguiar [32]. Dye decolorization percentage was calculated according to Equation (3):(3)$$Dye\;decolorization\;=\;\left(1-\frac{C_{t}}{C_{0}}\right)\;\times \;100\%$$
where *C*_0_ and *C_t_* are concentration values of dye at initial and a given time *t*, respectively. Residual concentrations of dyes were determined using calibration curves.

### 2.3. Kinetic Models

With respect to decolorization data, a kinetic study was carried out using zero-, first-, and second-order models, and another one which was developed by Behnajady et al.—BMG [34]. Zero-, first-, and second-order models were defined according to Equations (4)–(6) [35], respectively:(4)$$\frac{dC_{t}}{dt}\;=\;-k_{0}$$
(5)$$\frac{dC_{t}}{dt}\;=\;-k_{1}\cdot C_{t}$$
(6)$$\frac{dC_{t}}{dt}=\;-k_{2}\cdot {\left(C_{t}\right)}^{2}$$
where *k*_0_, *k*_1_, and *k*_2_ are apparent kinetic rate constants of zero-, first-, and second-order models, respectively, *t* is reaction time, and *C_t_* is dye concentration at a given time *t*. By integrating Equations (4)–(6), the following equations (Equations (7)–(9)) are obtained, respectively:(7)$$C_{t}\;=\;C_{0}-k_{0}\cdot t$$
(8)$$C_{t}\;=\;C_{0}.\exp\left(-k_{1}\cdot t\right)$$
(9)$$\frac{1}{C_{t}}\;=\;\frac{1}{C_{0}}\;+\;k_{2}\cdot t$$

Equation (8) is written in a linearized form as Equation (10):(10)$$\ln Ct\;=\;\ln Co-k_{1}\cdot t$$

The BMG model is expressed as Equation (11):(11)$$\frac{C_{t}}{C_{0}}\;=\;1-\left[\frac{t}{\left(m\;+\;b\cdot t\right)}\right]$$

Equation (11) is written in a linearized form as Equation (12):(12)$$\frac{t}{\left[1-\left(\frac{C_{t}}{C_{0}}\right)\right]}\;=\;m\;+\;b\cdot t$$
where *m* and *b* are two constants concerning initial reaction rate and maximum oxidation capacity, respectively [30].

### 2.4. Effect of Temperature

The effect of temperature on phenol red decolorization was evaluated at 20, 30, 40, and 50 °C. The concentrations of the reagents were those described in Section 2.2, except H_2_O_2_ (300 μmol L^−1^). In this set, experiments were carried out in triplicate and absorbance values were acquired at 3, 6, 9, 20, 30, 40, 50, and 60 min. According to the apparent first-order kinetic rate constants at different temperatures, apparent activation energy for phenol red decolorization was calculated according to Arrhenius equation [35]:(13)$$k=A\cdot \exp\left(-\frac{Ea}{R\cdot T}\right)$$
where *A* is a pre-exponential factor (or frequency; min^−1^); *Ea* is the apparent activation energy (J mol^−1^); *R* is the ideal gas constant (8.314 J mol^−1^ K^−1^); and T is absolute temperature (K). Equation (13) can be written in its linearized form as Equation (14):(14)$$\ln k=\ln\;A-\frac{E_{a}}{R\cdot T}$$

The Arrhenius plot of ln *k* versus 1/*T* allowed calculation of apparent activation energy values from the straight line slopes for the four reaction systems under evaluation.

## 3. Results and Discussion

### 3.1. Kinetic Evaluation of the Oxidative Decolorization of Different Dyes

Based on zero-, first-, and second-order reaction kinetics and the BMG model, Figure 2 shows the decolorization of safranin T dye by Fenton reactions, in both the absence and presence of 3-HAA. For zero- (Figure 2a), first- (Figure 2b), and second-order reaction kinetics (Figure 2c), the presence of 3-HAA steepened the slope of the curves, which indicates that there was an increase in the values of *k*_0_, *k*_1_, and *k*_2_, respectively, i.e., the decolorization rate of safranin T was increased by its presence.

Figure 2d shows the evaluation of decolorization data using the BMG. In this model, a plot of *t*/(1 − C/C_0_) versus *t* results in a straight line with an intercept *m* and a slope *b*. The physical meanings of the two constants can be illustrated by manipulating Equation (11), and substituting them for *t* in two extreme situations: (1) at the beginning of the reaction (where *t* = 0) and (2) towards the end of the reaction (where *t* is very long) [36]. This leads to Equations (15) and (16), respectively:

(15)$$\frac{dC}{dt}=-C_{0}\left(\frac{1}{m}\right)\;\mathrm{at}\;t=0$$

(16)$$C=C_{0}(1-\frac{1}{b})\;\mathrm{at}\;\mathrm{very}\;\mathrm{large}\;t$$

By comparing Equation (15) to the design equation of a constant-density batch reactor at *t* = 0 (i.e., *dC/dT│_t = 0_ = r*_0_), it is revealed that 1/*m* is related to its initial degradation rate (*−r*_0_), more precisely *−r*_0_ = C_0_(1/*m*). On the other hand, by comparing Equation (16) to the definition of conversion in a constant-density batch reactor, which is *X =* (*C*_0_ − *C*)/*C* or *C = C*_0_ (1 − *X*), it can be found that 1/*b* represents the maximum theoretical conversion that could have been achieved after a very long or infinite reaction time, hence named as the maximum oxidation capacity [34,36]. In Figure 2d, it is possible to observe that there was a decrease in the curve slope in the presence of 3-HAA. This means that Fe^2+^/H_2_O_2_/3-HAA obtained a higher oxidation capacity than Fe^2+^/H_2_O_2_. In short, 3-HAA displayed pro-oxidant properties by using either Fe^3+^ or Fe^2+^. A mechanism that is probably accountable for its pro-oxidant effect may be its Fe^3+^-reducing activity [31].

Linear regression analyses based on zero-, first-, and second-order reaction kinetics for the decolorization of dyes through Fenton processes were conducted so as to obtain the values of *k*_0_, *k*_1_, and *k*_2_, for which the results are shown in Table 2. Kinetic data on the BMG model were also obtained.

According to the correlation coefficient values of reactions containing Fe^2+^, the BMG kinetic model fitted better to the experimental data, as it obtained higher *R*^2^ values than zero-, first-, and second-order kinetic models. The oxidation of organic compounds by Fenton processes using Fe^2+^ as a catalyst generally proceeds via two stages: a fast one and a much slower one. The fast stage is attributed to a reaction between Fe^2+^ and H_2_O_2_, while the slower one is due to the accumulation of Fe^3+^ and a low recovery of Fe^2+^ by H_2_O_2_ [18,37]. This behavior was observed in a previous work for all dyes, mainly methyl orange, phenol red, and safranin T [32]. Chu et al. [38] observed the same performance for the degradation of 2,4-dichlorophenoxyacetic acid by Fe^2+^/H_2_O_2_. The reactions showed a two-stage pattern, i.e., a very fast stage followed by a slow one, hence they could not be modeled by zero-, first-, or second-order reaction kinetics. Therefore, the BMG model was more suitable to represent the Fe^2+^/H_2_O_2_ system kinetics. In previous studies, the fitting of experimental kinetic data on two-stage decolorization of several dyes by classical Fenton reaction was better with the BMG model [34,39,40,41,42,43].

By considering parameter 1/*b* obtained by the BMG model, it is possible to observe that the maximum oxidation capacity of Fe^2+^-containing reactions (*R*^2^ > 0.9) increased in the presence of 3-HAA, e.g., the oxidation capacity of safranin T was almost three times greater in the presence of 3-HAA. Santana et al. [42,43] showed the same upward trend in 1/*b* on the decolorization kinetics of different dyes by Fenton systems using dihydroxybenzenes and gallic acid as redox mediators. Regarding parameter 1/*m*, there were no significant changes in the initial degradation rates of dyes in the presence of 3-HAA. Except for the decolorization of methylene blue (*R*^2^ = 0.9017) and chromotrope 2R (*R*^2^ = 0.9612) using Fe^3+^/H_2_O_2_/3-HAA, the correlation coefficient values of Fe^3+^-containing reactions are not close to 1 (*R*^2^ < 0.9). Thus, when using Fe^3+^ as catalyst, there was no fitting of experimental data when using the BMG model. As for methylene blue and chromotrope 2R, the decolorization data were also well fitted to the second-order kinetic model with respect to reactions containing Fe^2+^. In this case, there was an increase in *k*^2^ values due to the presence of 3-HAA.

In general, the most appropriate model to describe Fe^3+^-containing reactions was the second-order reaction kinetics (*R*^2^ > 0.9), followed by first-order kinetics. Unlike Fe^2+^-containing reactions, these generally presented a single decolorization stage [32]. As for the decolorization of all dyes, the apparent kinetic rate constants of the second-order model, *k*_2_, considerably increased in the presence of 3-HAA. In the case of methylene blue and chromotrope 2R, *k*_2_ values were ten times and eight times greater, respectively, when using 3-HAA as a redox mediator. Concerning methyl orange, phenol red, and safranin T, the Fenton-like reaction barely decolorized these dyes (decolorization lower than 3%), and thus *k*_2_ presented very low values. Interestingly enough, the fungal metabolite 3-HAA showed a notable pro-oxidant activity in the degradation of these dyes using Fe^3+^ as a catalyst, since decolorization reached values close to 50%. In previous studies, there was also an effective decolorization of several dyes by Fenton-like reactions only in the presence of different redox mediators, such as dihydroxybenzenes and gallic acid [18], cysteine [21], and lignin-derived methoxyphenols [24].

By taking zero- and first-order reaction kinetics into account, it is possible to observe that they are also suitable for describing the decolorization of methylene blue by Fe^3+^/H_2_O_2_/3-HAA, phenol red by Fe^3+^/H_2_O_2_, chromotrope 2R, methyl orange, and safranin T by Fe^3+^/H_2_O_2_, and Fe^3+^/H_2_O_2_/3-HAA, since *R*^2^ > 0.9 in all cases. Suitability of more than one kinetic model is also commonly found in literature [39,40,43,44,45,46].

There are studies that only report the use of first-order reaction kinetics, such as Nidheesh et al. [13], who point out that this model is generally chosen. However, some correlation coefficients of the curves that describe the kinetics of reactions containing redox mediators for the first-order kinetic model are lower than 0.9 [20,22,33]. Three different kinetic models were analyzed in the decolorization of methyl orange by Fe^2+^/H_2_O_2_ by Youssef et al. [47], who found that none of them obtained a correlation coefficient higher than 0.9. It is very important to carry out an analysis based on different models for a more precise determination of kinetic parameters; this makes it possible to find the most suitable reactor configuration for degrading an organic pollutant by Fenton processes. In addition, it is important to highlight the data presented by Xu et al. [45,46,48]. They studied different dosages of reagents, catalysts, and target dyes, and observed that the most suitable kinetic model can change from one reaction system to another. This fact was also verified through the results found herein.

The effect of 3-HAA concentration on phenol red decolorization, which has also been studied previously [32], was also evaluated using kinetic models, as shown in Table 3. According to the average *R*^2^ values, second-order reaction kinetics were the most adequate to describe these reactions, followed by the first-order kinetic model. It is possible to observe that an increase in 3-HAA concentration from 10 to 30 mmol L^−1^ led to *k*_2_ values ranging from 0.0005 to 0.0022 L µmol^−1^ min^−1^. The reactions performed at 60 and 90 mmol L^−1^ had a scarcely increased *k*_2_ (in comparison with 30 mmol L^−1^ of this mediator). On the other hand, 3-HAA concentrations of over 120 mmol L^−1^ sharply decreased both phenol red decolorization and *k*_2_ values, due to the scavenging effect of the redox mediator at higher concentrations. There was a similar trend between the decolorization data (after 60 min) and *k*_2_ values as a function of 3-HAA concentration. Excessive 3-HAA concentrations can compete with its target substrate for HO^●^ generation, thus impairing phenol red decolorization. Pro-oxidant properties in Fenton processes are generally observed at low concentrations of redox mediators, as opposed to antioxidant properties [17,19,23,25]. This aspect is desirable, since the elevated cost of such mediators would make their use infeasible. The use of a low mediator concentration is also important for generating fewer byproducts, which can also be attacked by HO^•^ radicals and converted into lower molecular mass compounds, including CO_2_ and water [19].

### 3.2. Effect of Temperature

Reaction temperature is a relevant operating parameter in Fenton processes [5,13]. Therefore, the effect of reaction temperature on phenol red decolorization was evaluated to calculate the apparent activation energy. Phenol red was one of the most recalcitrant dyes in our previous work [32]. Thus, it was also chosen for studies involving different temperatures. As shown in Figure 3a–d, an increase in temperature favored decolorization in the four systems under evaluation, in the following order: Fe^2+^/H_2_O_2_/3-HAA > Fe^2+^/H_2_O_2_ > Fe^3+^/H_2_O_2_/3-HAA > Fe^3+^/H_2_O_2_. This increase is due to higher temperatures, which increased the reaction rate between hydrogen peroxide and iron, thus increasing the generation of reactive oxygen species, such as the HO^●^ radical. These results also imply that such reactions are endothermic. At higher temperatures, the time length required for decolorization was much shorter than at lower temperatures. A complete decolorization of phenol red was observed in the Fe^2+^/H_2_O_2_/3-HAA system, regardless of temperature conditions. As for the other systems, a complete decolorization was achieved after 60 min of reaction carried out at 50 °C. Through the Fe^2+^/H_2_O_2_ system, decolorization was incomplete only at 20 °C.

The reactions performed at different temperatures were also tested using the four kinetic models, and data on the kinetic parameters are shown in Table 4. With regard to the three classical kinetic models, it was possible to verify an increase in apparent rate constants as a function of temperature, and also due to adding 3-HAA by considering the same temperature. It is important to highlight that the models did not fit all reactions well. With respect to the BMG model, it was possible to observe increased oxidation capacity and a faster initial reaction rate due to higher temperatures and the addition of 3-HAA. Oxidation capacity was maximum (1/*b* ~1) for all reactions evaluated at 50 °C, and also for the Fe^2+^/H_2_O_2_/3-HAA system. In this set of experiments, the BMG model also presented better goodness of fit to decolorization data on reactions containing Fe^2+^. As regards the Fe^3+^/H_2_O_2_/3-HAA system, a two-stage decolorization was also observed, thus demonstrating goodness of fit by the BMG model. These data show that the Fenton oxidative process for phenol red decolorization was faster with the addition of 3-HAA, thus indicating that it has pro-oxidant properties at different temperatures.

In Figure 4a, it is possible to observe an increase in *k*_1_ values as a function of temperature for the four systems being tested. The kinetics modeling for this set of experiments showed that decolorization kinetics achieved better goodness of fit through the first-order model (average *R*^2^ = 0.9562). Next, ln *k*_1_ vs. *1/T* plots, according to Arrhenius’ law, were obtained in order to determine the values of apparent activation energy for phenol red decolorization (Figure 4b). A good linear relationship was observed in the ln *k*_1_ vs. *1/T* plots for all reaction systems (*R*^2^ > 0.9). Due to adding 3-HAA, apparent activation energy decreased from 63.7 to 45.8 kJ mol^−1^ and from 106.8 to 64.8 kJ mol^−1^ for reactions containing Fe^2+^ and Fe^3+^, respectively. By comparing the two oxidation states of iron, apparent activation energy with Fe^2+^ was lower, a result that was expected since there is greater generation of HO^●^ from Fe^2+^ [5,14]. Lower activation energy values show that an oxidative reaction proceeds with a low energy barrier [49]. Generally, activation energy of ordinary thermal reactions ranges between 60 and 250 kJ mol^−1^ [50]. The present results demonstrate that phenol red decolorization through Fenton processes can be easily achieved, moreover, it can be improved in the presence of 3-HAA. Luo et al. [21] also observed a decrease in activation energy due the presence of another reducer, i.e., cysteine, for methylene blue decolorization by Fe^3+^/H_2_O_2_. In the present work, 3-HAA not only allowed a decrease of the activation energy of Fe^3+^/H_2_O_2_, but also that of the Fe^2+^/H_2_O_2_.

Fenton processes are performed in four stages: pH adjustment to low acidic values (when necessary), oxidation reaction, neutralization, and coagulation [13]. Beyond the former, the neutralization step also enhances the cost of treatment. However, the third step permits coagulation of the iron at high values of pH, allowing the desirable and simultaneous removal of the metal and other coagulated residual substances. Thus, iron ions have dual functions of oxidation and coagulation [51]. These processes are promising in degrading recalcitrant organic pollutants, but more studies involving redox mediators still need be performed. It is important to emphasize that the addition of the redox mediator is a complex approach to treating industrial effluents. As previously suggested by Santana and Aguiar [32], the addition of 3-HAA to Fenton processes is far from practical, due to its elevated cost. Recently, the use of aqueous wood extracts [52] or an industrial effluent [26], both containing natural compounds, have improved the degradation of different organic pollutants through Fenton processes. These solutions could serve as low-cost alternatives for obtaining redox mediators. In addition, the toxicity and biodegradability of these compounds should also be considered, especially if the treated effluent will be discarded in water bodies and if it still contains some of these aforementioned mediators. In this sense, it is worth mentioning that, when reacting with Fe^3+^, they are degraded and even mineralized [14], which is desirable for real applications.

## 4. Conclusions

In the present work, a Fe^3+^ reducer produced by fungi, i.e., 3-hydroxyanthranilic acid, demonstrated improvements in kinetic parameters (apparent rate constants and maximum oxidative capacity) of reactions containing either Fe^2+^ or Fe^3+^ as catalysts to decolorize different dyes in Fenton processes. Fe^3+^-containing reactions were well fitted with first-order and mainly second-order kinetic models, whereas the BMG model was the most appropriate with respect to Fe^2+^. These aspects demonstrate that 3-HAA can accelerate decolorization reactions of dyes. In addition, a decrease in apparent activation energy was observed due to the addition of 3-HAA, thus demonstrating an important pro-oxidant property of this redox mediator through Fenton oxidative processes.

## Figures and Tables

**Figure 1 ijerph-16-01602-f001:**
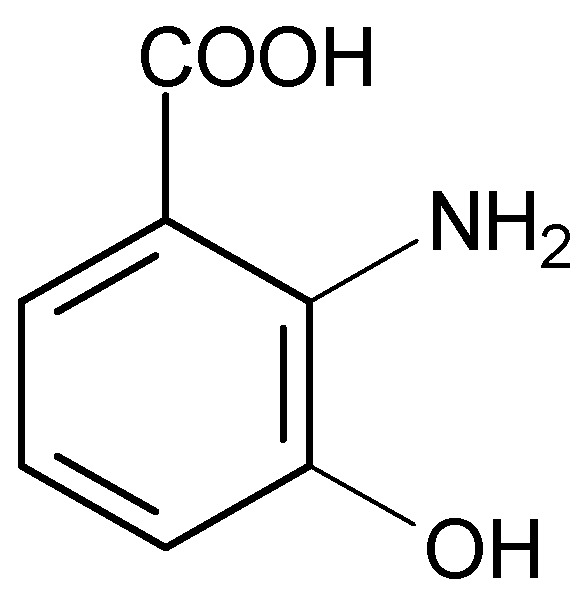
Chemical structure of 3-hydroxyanthranilic acid (3-HAA).

**Figure 2 ijerph-16-01602-f002:**
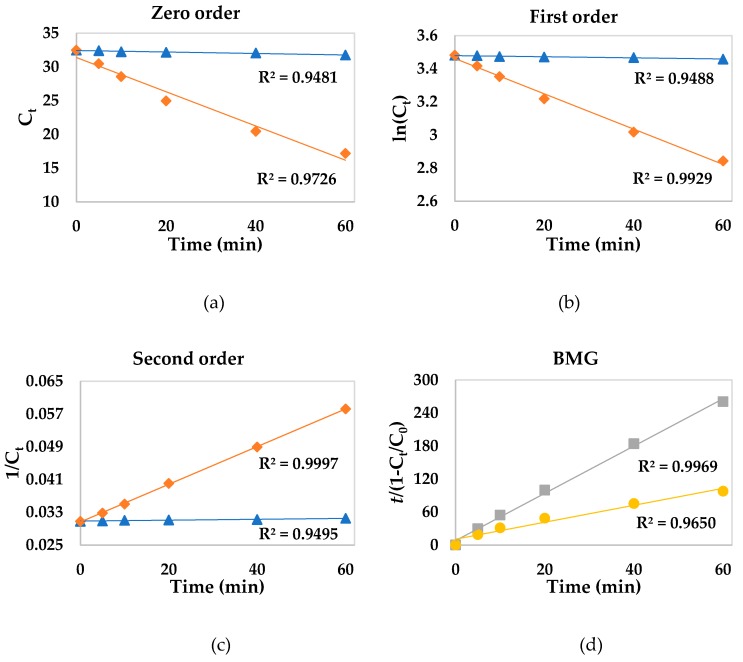
Decolorization data of safranin T dye using different kinetic models. Reaction systems: Fe^3+^/H_2_O_2_ (▲), Fe^3+^/H_2_O_2_/3-HAA (◆), Fe^2+^/H_2_O_2_ (■), and Fe^2+^/H_2_O_2_/3-HAA (●). [Fe] = 30 μmol L^−1^; [H_2_O_2_] = 450 μmol L^−1^; [safranin T] = 30 μmol L^−1^; [3-HAA] = 10 μmol L^−1^; pH = 2.5–3.0.

**Figure 3 ijerph-16-01602-f003:**
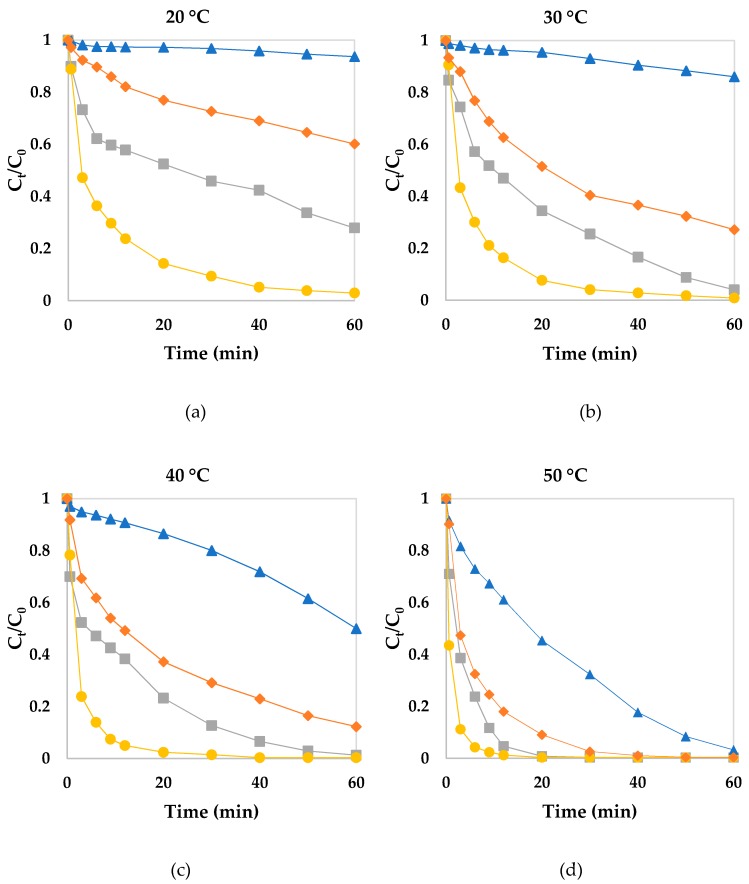
Effect of temperature on phenol red decolorization through different Fenton systems. (▲), Fe^3+^/H_2_O_2_/3-HAA (◆), Fe^2+^/H_2_O_2_ (■), and Fe^2+^/H_2_O_2_/3-HAA (●). [Fe] = 30 μmol L^−1^; [H_2_O_2_] = 300 μmol L^−1^; [phenol red] = 30 μmol L^−1^; [3-HAA] = 10 μmol L^−1^; pH = 2.5–3.0.

**Figure 4 ijerph-16-01602-f004:**
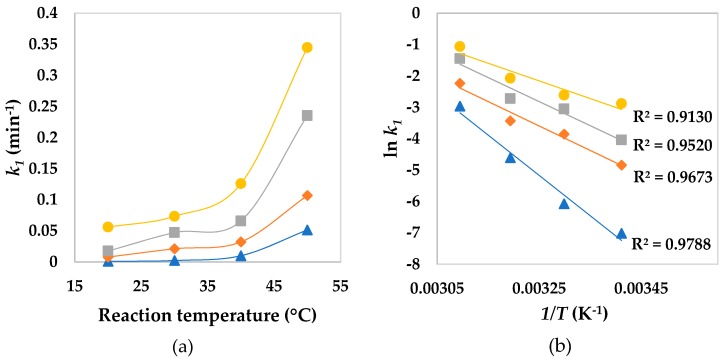
Effect of temperature on *k*_1_ of phenol red decolorization through Fenton processes (**a**); Arrhenius plots of phenol red decolorization at different temperatures (**b**): Fe^3+^/H_2_O_2_ (▲), Fe^3+^/H_2_O_2_/3-HAA (◆), Fe^2+^/H_2_O_2_ (■), and Fe^2+^/H_2_O_2_/3-HAA (●). [Fe] = 30 μmol L^−1^; [H_2_O_2_] = 300 μmol L^−1^; [phenol red] = 30 μmol L^−1^; [3-HAA] = 10 μmol L^−1^; pH = 2.5–3.0.

**Table 1 ijerph-16-01602-t001:** Features of dyes evaluated in this study.

Dyes	Type	λ_max_ (nm)	Color Index	Chemical Structure
Methylene blue	Thiazine	665	52015	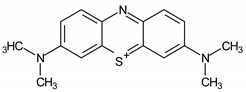
Chromotrope 2R	Azo	513	16570	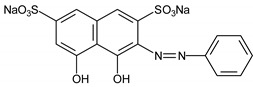
Methyl orange	Azo	508	13025	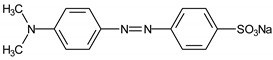
Phenol red	Triphenylmethane	435	-	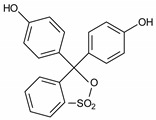
Safranin T	Thiazine	519	50240	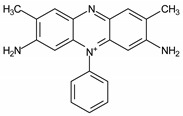

**Table 2 ijerph-16-01602-t002:** Dye decolorization percentages after 60 min through Fenton processes, apparent kinetic rate constants of the zero- (*k*_0_), first- (*k*_1_), and second-order(*k*_2_), parameters obtained on the BMG model (1/*m* and 1/*b*) and correlation coefficients (*R*^2^) obtained after data fits.

Dye	Reaction Systems	Decolorization (%)	Zero Order		First Order		Second Order		BMG		
			*k*_0_ (µmol L^−1^ min^−1^)	*R* ^2^	*k*_1_ (min^−1^)	*R* ^2^	*k*_2_ (L µmol^−1^ min^−1^)	*R* ^2^	1/*m* (min^−1^)	1/*b*	*R* ^2^
**Methylene Blue**	Fe^2+^	66.7 ± 0.2	0.2958	0.7226	0.0153	0.8686	0.0009	0.9587	0.1343	0.6872	0.9765
	Fe^2+^/3-HAA	87.0 ± 0.3	0.3497	0.7889	0.0289	0.9642	0.0030	0.9816	0.1401	0.9014	0.9711
	Fe^3+^	31.3 ± 2.0	0.1432	0.8754	0.0050	0.8881	0.0002	0.8912	0.0239	0.3036	0.7689
	Fe^3+^/3-HAA	81.3 ± 0.0	0.4108	0.9199	0.0264	0.9914	0.0020	0.9502	0.0744	0.9069	0.9017
**Chromotrope 2R**	Fe^2+^	72.6 ± 0.6	0.2381	0.5808	0.0171	0.7673	0.0014	0.9180	0.2453	0.7420	0.9926
	Fe^2+^/3-HAA	80.3 ± 0.2	0.2781	0.6302	0.0234	0.8451	0.0023	0.9729	0.2382	0.8338	0.9919
	Fe^3+^	27.0 ± 0.1	0.1043	0.9058	0.0045	0.9287	0.0002	0.9450	0.0243	0.2821	0.8648
	Fe^3+^/3-HAA	73.7 ± 1.2	0.2951	0.8344	0.0208	0.9521	0.0016	0.9979	0.1000	0.8013	0.9612
**Methyl Orange**	Fe^2+^	50.3 ± 0.2	0.1751	0.5677	0.0087	0.6664	0.0004	0.7655	0.1739	0.5110	0.9920
	Fe^2+^/3-HAA	58.7 ± 1.7	0.2355	0.7145	0.0123	0.8364	0.0007	0.9283	0.1233	0.6100	0.9799
	Fe^3+^	2.7 ± 0.1	0.0128	0.9721	0.0004	0.9733	0.00001	0.9745	0.0017	0.0326	0.7756
	Fe^3+^/3-HAA	44.4 ± 0.6	0.2321	0.9654	0.0099	0.9871	0.0004	0.9980	0.0184	0.6811	0.5385
**Phenol red**	Fe^2+^	41.1 ± 0.1	0.1331	0.3345	0.0052	0.3675	0.0002	0.4066	0.4852	0.4124	0.9993
	Fe^2+^/3-HAA	58.9 ± 0.3	0.2303	0.5576	0.0112	0.6863	0.0006	0.8119	0.2111	0.6002	0.9929
	Fe^3+^	2.9 ± 0.1	0.0169	0.9548	0.0005	0.9562	0.00001	0.9575	0.0010	0.0517	0.3551
	Fe^3+^/3-HAA	48.8 ± 1.4	0.2672	0.8332	0.0111	0.8890	0.0005	0.9353	0.0306	0.6685	0.6571
**Safranin T**	Fe^2+^	23.0 ± 0.1	0.0809	0.4651	0.0029	0.4942	0.0001	0.5251	0.1235	0.2328	0.9969
	Fe^2+^/3-HAA	61.4 ± 0.6	0.2737	0.8073	0.0139	0.9132	0.0008	0.9757	0.0939	0.6491	0.9650
	Fe^3+^	2.3 ± 0.7	0.0111	0.9481	0.0003	0.9488	0.00001	0.9495	0.0010	0.0279	0.6178
	Fe^3+^/3-HAA	47.1 ± 1.5	0.2537	0.9726	0.0107	0.9929	0.0005	0.9997	0.0211	0.6727	0.6279

**Table 3 ijerph-16-01602-t003:** Decolorization data of phenol red by Fe^3+^/H_2_O_2_ performed at varied concentrations of 3-HAA, apparent kinetic rate constants of the zero- (*k*_0_), first- (*k*_1_), and second-order (*k*_2_), parameters obtained using the BMG model (1/*m* and 1/*b*), and correlation coefficients (*R*^2^) obtained after data fits.

3-HAA (µmol L^−1^)	Decolorization (%)	Zero Order		First Order		Second Order		BMG		
		*k*_0_ (µmol L^−1^ min^−1^)	*R* ^2^	*k*_1_ (min^−1^)	*R* ^2^	*k*_2_ (L µmol^−1^ min^−1^)	*R* ^2^	1/*m* (min^−1^)	1/*b*	*R* ^2^
0	2.9 ± 0.1	0.0169	0.9548	0.0005	0.9562	0.00001	0.9575	0.0010	0.0517	0.3551
10	48.8 ± 1.4	0.2672	0.8332	0.0111	0.8890	0.0005	0.9353	0.0306	0.6685	0.6571
30	77.5 ± 0.7	0.3674	0.7692	0.0231	0.9199	0.0017	0.9931	0.0870	0.9045	0.9780
60	81.8 ± 0.2	0.,4173	0.8464	0.0277	0.9693	0.0022	0.9982	0.0618	1.0514	0.9494
90	81.1 ± 0.1	0.4046	0.8315	0.0266	0.9643	0.0021	0.9983	0.0675	1.0085	0.9591
120	71.1 ± 0.4	0.3646	0.8836	0.0199	0.9708	0.0012	0.9992	0.0467	0.9395	0.9322
150	63.3 ± 0.1	0.3513	0.9410	0.0169	0.9845	0.0009	0.9989	0.0247	1.1323	0.7891
180	50.3 ± 0.4	0.2778	0.9741	0.0116	0.9951	0.0005	0.9994	0.0247	1.1323	0.7109
**Average *R*^2^**			0.8792		0.9561		0.9889			0.8537

**Table 4 ijerph-16-01602-t004:** Decolorization data of phenol red at different temperatures, apparent kinetic rate constants of the zero- (*k*_0_), first- (*k*_1_), and second-order (*k*_2_), parameters obtained on the BMG model (1/*m* and 1/*b*) and correlation coefficients (*R*^2^) obtained after data fits.

Reaction Systems	Temperature (°C)	Decolorization (%)	Zero Order		First Order		Second Order		BMG		
			*k*_0_ (µmol L^−1^ min^−1^)	*R* ^2^	*k*_1_ (min^−1^)	*R* ^2^	*k*_2_ (L µmol^−1^ min^−1^)	*R* ^2^	1/*m* (min^−1^)	1/*b*	*R* ^2^
**Fe^3+^/H_2_O_2_**	20	6.3 ± 0.6	0.0262	0.8870	0.0009	0.8911	0.00003	0.8949	0.0052	0.0619	0.8376
	30	13.9 ± 0.6	0.0718	0.9874	0.0023	0.9875	0.00007	0.9867	0.0065	0.1591	0.6554
	40	50.0 ± 1.4	0.2729	0.9803	0.0101	0.9510	0.0004	0.9055	0.0148	0.6343	0.4193
	50	96.7 ± 0.3	0.4049	0.5946	0.0451	0.9776	0.0047	0.8011	0.0793	1.0513	0.8707
**Fe^3+^/H_2_O_2_/3-HAA**	20	39.9 ± 0.8	0.2266	0.9342	0.0079	0.9633	0.0003	0.9827	0.0316	0.4448	0.9156
	30	72.8 ± 0.5	0.4250	0.8779	0.0212	0.9604	0.0012	0.9952	0.0706	0.8335	0.9461
	40	87.7 ± 0.5	0.3615	0.8181	0.0323	0.9741	0.0037	0.9612	0.1308	0.9354	0.9789
	50	99.6 ± 0.0	0.6406	0.9569	0.1068	0.9910	0.1503	0.7565	0.4258	1.0427	0.9964
**Fe^2+^/H_2_O_2_**	20	72.1 ± 0.9	0.3470	0.7909	0.0176	0.9139	0.0010	0.9567	0.1220	0.7224	0.9695
	30	96.0 ± 0.3	0.4685	0.8518	0.0473	0.9802	0.0087	0.7557	0.1359	1.0192	0.9695
	40	98.7 ± 0.6	0.4724	0.7577	0.0659	0.9894	0.0255	0.7349	0.2304	1.0357	0.9857
	50	99.7 ± 0.0	1.4234	0.7039	0.2354	0.9960	0.1660	0.7542	0.7482	1.0447	0.9889
**Fe^2+^/H_2_O_2_/3-HAA**	20	97.0 ± 0.1	0.4774	0.5941	0.0561	0.9408	0.0135	0.9640	0.3140	1.0180	0.9965
	30	99.1 ± 0.2	0.3654	0.5234	0.0734	0.9442	0.0500	0.8474	0.3703	1.0393	0.9958
	40	99.6 ± 0.0	0.4989	0.4227	0.1260	0.9076	0.1896	0.7599	1.4449	1.0099	0.9995
	50	99.7 ± 0.0	2.0143	0.5741	0.3448	0.9308	0.1911	0.9229	3.4037	1.0102	0.9996
**Average *R*^2^**				0.7659		0.9562		0.8737			0.9078

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
