# Peer review of "Kinetic Evaluation of Dye Decolorization by Fenton Processes in the Presence of 3-Hydroxyanthranilic Acid"

_ijerph, 2019, doi:10.3390/ijerph16091602_

Round 1

Reviewer 1 Report

The main objective of the study is to derive in the controlled experiments the oxidation constants of several dyes by Fenton reaction which is mediated by 3-hydroxyanthranilic acid (3-HAA). The principle that antioxidations 3-HAA can be used for reducing ferric iron to ferrous are known, whereas the kinetics has not been established yet. The overall idea is that if 3-HAA that is produced by white-rod fungi (in kynurenine pathway)can be used to in stead of adding acid (Fenton works only at low pH) this would make process cheeper thus more widely used. 

The quality of the study is good and the obtained data are  sufficient to support conclusions. However, there is need to elaborate in discussion part about possible scheme of technology which could encompass the principle in question. For example, what type of reactors can be used. 

Moreover. what are by products in which concentration of 3-HAA after reaction will be generated and how this would effect overall quality of water. 

Author Response

Responses to reviewer 1

We would like to emphasize that the reviewer comments significantly contributed to improve our manuscript. Our gratitude should be expressed to the reviewers for taking the time and effort to understand and improve the presentation of our work. A point-by-point response is provided, addressing all of the reviewer comments and suggestions. In the text, the main changes are highlighted.

Reviewer: The main objective of the study is to derive in the controlled experiments the oxidation constants of several dyes by Fenton reaction which is mediated by 3-hydroxyanthranilic acid (3-HAA). The principle that antioxidations 3-HAA can be used for reducing ferric iron to ferrous are known, whereas the kinetics has not been established yet. The overall idea is that if 3-HAA that is produced by white-rod fungi (in kynurenine pathway) can be used to instead of adding acid (Fenton works only at low pH) this would make process cheaper thus more widely used. The quality of the study is good and the obtained data are sufficient to support conclusions.

Authors: We thank the reviewer for this comment.

Reviewer: However, there is need to elaborate in discussion part about possible scheme of technology which could encompass the principle in question. For example, what type of reactors can be used. 

Authors:  The present work aimed to evaluate only the effect of 3-HAA as redox mediator in Fenton processes using different kinetic models. In further works, we intend to evaluate the optimization of these reactions in the presence of mediator and thereby suggest what type of reactor could be most appropriate.

Reviewer: Moreover. what are by products in which concentration of 3-HAA after reaction will be generated and how this would affect overall quality of water.

Authors:  The products of the reaction between 3-HAA and Fenton reagents are still known. We inserted now the following sentence in lines 267-270: “The use of low mediator concentration is also important to generate less its by-products, which can also be attacked by HO• radicals and converted into lower molecular weight compounds, including CO2 and water [19].”

Reviewer 2 Report

General Comments

The work show that 3-hydroxyanthranilic acid (3-HAA) can act as a redox mediator in the Fenton and Fenton-like degradation of several dyes (methylene blue, chromotrope 2R, methyl orange, phenol red, and safranin T). The discoloration of each dye is monitored by UV-visible spectroscopy and represented in different ways for comparison of zero order, first order, second order, and BMG kinetic models. In addition, activation energy for the degradation of phenol red for the system under study is reported. Overall, this is a comprehensive piece of work that should be published after a minor revision to address of couple of very important issues.

Major Comments

1)      The experimental procedure uses pH adjustment to 2.5-3 (line 102). The manuscript needs to carefully address this issue by explaining how such a low pH can be provided to the wastewaters of interest. After generating the acidic water, the manuscript also needs to introduce a way to solve the new low pH waste/problem created. In addition, it would be important for the manuscript to discuss what the cost of applying such technology could be after accounting for dealing with the acidic waste.

2)      It would be important that the introduction also covers some recent references about effective ways to degrade recalcitrant organics in water and at interfaces following the statement in lines 39-40. The manuscript should mention the importance of ozone treatment and irradiation to degrade aromatics and cite the following five papers:

a)      Pillar et al., Environ. Sci. Technol., 2017, 51 (9), pp 4951–4959.

b)      Okasaki et al., Appl. Mag. Resonance, 2018, 49 (8), pp 881–892.

c)      Albarran et al., Rad. Phys. Chem. 2018, 147, pp 27-34.

d)      Eugene et al., Molecules 2019, 24 (6), 1124.

e)      Albarran, et al., Rad. Phys. Chem., 2016, 124, pp 46-51.

Minor Comments

1)      Remove the bold font from references 5, 9, 15, 18, 27, and 41.

2)      Remove the solid green background from Figures 2, 3, and 4.

3)      Define BMG in the abstract (line 20), and do so the first time mentioned in the text too.

Author Response

Responses to reviewer 2

Authors: We would like to emphasize that the reviewer comments significantly contributed to improve our manuscript. Our gratitude should be expressed to the reviewers for taking the time and effort to understand and improve the presentation of our work. A point-by-point response is provided, addressing all of the reviewer comments and suggestions. In the text, the main changes are highlighted.

Reviewer: The work show that 3-hydroxyanthranilic acid (3-HAA) can act as a redox mediator in the Fenton and Fenton-like degradation of several dyes (methylene blue, chromotrope 2R, methyl orange, phenol red, and safranin T). The discoloration of each dye is monitored by UV-visible spectroscopy and represented in different ways for comparison of zero order, first order, second order, and BMG kinetic models. In addition, activation energy for the degradation of phenol red for the system under study is reported. Overall, this is a comprehensive piece of work that should be published after a minor revision to address of couple of very important issues.

Authors: We thank the reviewer for this comment.

Major Comments

1) The experimental procedure uses pH adjustment to 2.5-3 (line 102). The manuscript needs to carefully address this issue by explaining how such a low pH can be provided to the wastewaters of interest. After generating the acidic water, the manuscript also needs to introduce a way to solve the new low pH waste/problem created. In addition, it would be important for the manuscript to discuss what the cost of applying such technology could be after accounting for dealing with the acidic waste.

Authors: We inserted now the following sentence in lines 336-342: “Fenton processes are performed in four stages: pH adjustment to low acidic values (when necessary), oxidation reaction, neutralization, and coagulation [13]. Beyond the former, the neutralization step also enhancing the cost of treatment. However, this step permits the coagulation effect of the iron at higher values of pH, allowing the desirable and simultaneous removal of the metal and other coagulated residual substances. Thus, iron ions have dual functions of oxidation and coagulation [51]. These processes are promising in degrading recalcitrant organic pollutants, but more studies involving redox mediators still need be performed.”

2) It would be important that the introduction also covers some recent references about effective ways to degrade recalcitrant organics in water and at interfaces following the statement in lines 39-40. The manuscript should mention the importance of ozone treatment and irradiation to degrade aromatics and cite the following five papers:

a)      Pillar et al., Environ. Sci. Technol., 2017, 51 (9), pp 4951–4959.

b)      Okasaki et al., Appl. Mag. Resonance, 2018, 49 (8), pp 881–892.

c)      Albarran et al., Rad. Phys. Chem. 2018, 147, pp 27-34.

d)      Eugene et al., Molecules 2019, 24 (6), 1124.

e)      Albarran, et al., Rad. Phys. Chem., 2016, 124, pp 46-51.

Authors: We inserted now the references about irradiation (Albarran et al 2016 and 2018) and ozone treatment (Pillar et al 2017; Tang et al., Environ. Sci. Poll. Res. 2016, 23:18800–18808) in line 40.

Minor Comments

1)      Remove the bold font from references 5, 9, 15, 18, 27, and 41.

Authors: corrected.                                                         

2)      Remove the solid green background from Figures 2, 3, and 4.

Authors: The green color is only in the pdf version.           

3)      Define BMG in the abstract (line 20), and do so the first time mentioned in the text too.

Authors: corrected.                                                         
